# Whole-body sensorimotor skill learning in football players: No evidence for motor transfer effects

**Tom Maudrich**[1,2]*, **Hannah Kandt**[1], **Patrick Ragert**[1,2], **Rouven Kenville**[1,2]*

**1** Department of Movement Neuroscience, Faculty of Sport Science, Leipzig University, Leipzig, Saxony, Germany, **2** Department of Neurology, Max Planck Institute for Human Cognitive and Brain Sciences, Leipzig, Saxony, Germany

* tom.maudrich@uni-leipzig.de (TM); rouven.kenville@uni-leipzig.de (RK)

**Data Availability Statement:** The data underlying the results presented in the study are available from https://figshare.com/articles/dataset/Data_-_SRTT-Football/19948211.

## Abstract

Besides simple movement sequences, precise whole-body motor sequences are fundamental for top athletic performance. It has long been questioned whether athletes have an advantage when learning new whole-body motor sequences. In a previous study, we did not find any superior learning or transfer effects of strength and endurance athletes in a complex whole-body serial reaction time task (CWB-SRTT). In the present study, we aimed to extend this research by increasing the overlap of task requirements between CWB-SRTT and a specific sports discipline. For this purpose, we assessed differences between football players and non-athletes during motor sequence learning using CWB-SRTT. 15 non-athletes (CG) and 16 football players (FG) performed the CWB-SRTT over 2 days separated by one week. Median reaction times and movement times were analyzed as well as differences in sequence-specific CWB-SRTT learning rates and retention. Our findings did not reveal any differences in sequence-specific or non-sequence-specific improvement, nor retention rates between CG and FG. We speculate that this might relate to a predominately cognitive-induced learning effect during CWB-SRTT which negates the assumed motor advantage of the football players.

## Introduction

There is an ongoing debate about the extent to which specific athletic performance capabilities exhibited during participation in a given sport-related activity translate to superior motor performance in an unrelated motor scenario. Previous research indicates that such transfer effects (motor transfer) depend on the degree of similarity between the domain of athletic expertise and the new task to be performed [1, 2]. Such positive motor transfer has already been demonstrated in baseball players [1], Karate athletes [3], as well as basketball and field hockey players [4] during a variety of simple motor tasks. However, the effect is absent once the new task shows little to no overlap with the domain of athletic expertise [2, 5]. To date, few studies exist on motor transfer abilities of athletes in motor sequence learning. Notably, musicians demonstrate improved learning performance in related fine motor sequence and tapping tasks

**Funding:** The author(s) received no specific funding for this work.

**Competing interests:** The authors have declared that no competing interests exist.

compared to non-musicians [6, 7], potentially supporting the idea of expertise-induced motor transfer in such tasks.

Precise coordination of motor sequences plays a fundamental role in the process of acquiring and mastering everyday motor tasks as well as athletic performance. Basic motor skills such as typing a phone number, as well as the quality of complex athletic movements, depend on the ability to manifest specific motor sequences [8]. Motor sequence learning is typically investigated using serial reaction time tasks (SRTT) for upper or lower extremities. During SRTT, participants are presented with spatially separated visual stimuli. Participants must respond as accurately and as quickly as possible with a motor action, e.g., pressing a button, corresponding to the sequence of the presented stimuli. Performance of motor sequence learning is measured by the proportion of correct responses and the reaction time to the presented visual cues. Consequently, SRTT performance relies on the successful integration of motor and cognitive processes. SRTT designs combine sequence blocks (fixed order of stimuli) with randomized blocks (random order of stimuli) to investigate sequential and non-sequential parts during the motor learning process [9].

As an extension of the conventional SRTT, which is mainly applied in simple movement tasks, we introduced a complex whole-body SRTT (CWB-SRTT) [10]. Initial evidence demonstrated that the brains of non-athletes functionally reorganize during learning of CWB-SRTT [10]. A subsequent study examined baseline differences in initial performance during CWB-SRTT as well as learning rates between athletes (endurance and strength athletes) and non-athletes during two days of CWB-SRTT training [11]. The results of this study did not show any differences in initial performance or learning rates between athletes and non-athletes. Accordingly, a transfer effect of basic motor abilities from strength and endurance athletes on CWB-SRTT performance could not be demonstrated. The task specificity of the CWB-SRTT was pointed out as a possible explanation for these results. Although endurance and strength athletes internalize specific movement patterns within their many years of training, these patterns are distinguishable from the typical motor actions of the CWB-SRTT. This discrepancy between the typical motor actions of strength and endurance athletes and the necessary motor actions within the CWB-SRTT possibly accounts for the lack of motor transfer effects.

To more accurately assess the possibility of positive motor transfer on motor sequence learning, the intersection between the movement patterns of athletes and those required in the CWB-SRTT must be expanded. Fundamentally, the CWB-SRTT places demands on the serial reaction ability of the lower extremities in a whole-body compound movement. Athletes from several sports can be considered as potential study populations, however, based on their requirement profiles, the largest motor-related intersection lies with football. In football, offensive, as well as defensive game decisions, are realized in the range of a few hundredths of a second via motor actions of the lower extremities [12]. Accordingly, reaction time is an important determinant of football performance [12]. Several studies illustrate this by demonstrating that the cognitive processing of soccer players, in the form of improved reaction times during general motor inhibition tasks [13] as well as faster stimulus processing times and improved attentional performance [14], is increased compared to control groups. Furthermore, the motor related overlap between CWB-SRTT and football performance, i.e., the rapid, goal-directed activation of the lower extremities, is considerably high. In sum, football players might therefore hold an advantage in learning a novel sensorimotor task such as the CWB-SRTT.

Consequently, the purpose of this study was to examine the difference between football players and non-athletes during CWB-SRTT performance on two separate days to investigate potential motor transfer of football skills when learning a new motor skill using CWB-SRTT.

Based on the outlined body of research, we hypothesized improved initial performance as well as improved learning and retention rates for football players compared to non-athletes.

## Materials and methods

### Ethical approval

This study was supported by the local ethics committee of Leipzig University (ref. nr. 287/18-ek). According to the Declaration of Helsinki, all subjects provided written informed consent to participate in the study.

### Participants

A total of 31 participants (15 female, 16 male; age (mean ± standard deviation): 23.0 ± 2.6 years) were enrolled in the present study, recruited through a public advertisement based on the following inclusion criteria: age 18–35 years & neurologically healthy. Participants were separated into two groups according to their participation in organized football training: a non-athlete control group (CG; n = 15; age: 22.3 ± 2.6 years) and a football group (FG; n = 16; age: 23.6 ± 2.6 years). As non-athletes were considered those participants with an upper limit of 3 hrs of general physical exercise a week (2.0 ± 1.4 hrs). Non-athletes did not participate in a specific sport in an organized manner apart from general recreational activities (e.g., running, fitness and cycling). FG had to regularly undergo at least 3 organized football training sessions during an average week (6.3 ± 1.8 hrs). All football players in this study had to have participated in organized training for at least two years. This critical value was exceeded by FG (17.3 ± 3.9 years). Handedness and footedness of all participants were assessed using the Edinburgh Handedness Inventory [15] and the Waterloo Footedness Questionnaire Revised [16], respectively. An overview of demographic and laterality variables is presented in Table 1.

### Experimental procedure

Participants performed a whole-body sensorimotor skill task, the CWB-SRTT, on two separate days with one week of rest in between sessions, with the measurements of one participant taking place at comparable daytimes. FG and CG completed 15 consecutive sequence blocks and one random block before and one random block after all sequence blocks on each day (Fig 1A). This leads to a total of 204 motor responses for each experimental day. The CWB-SRTT lasted 15 minutes, with 15-second inter-block rest intervals.

**Table 1. Overview of participant characteristics.**

| Variable | Football group (FB) | Control group (CG) | Sign. Mann-Whitney |
|---|---|---|---|
| Sample size | n = 16 | n = 15 | - |
| Gender (male/female) | 8/8 | 8/7 | - |
| Age (years) | 23.56 ± 2.6 | 22.3 ± 2.6 | p = 0.134 |
| Handedness | 70.00 ± 42.4 | 76.00 ± 40.8 | p = 0.440 |
| Footedness | 5.44 ± 8.6 | 11.00 ± 7.1 | p = 0.065 |
| Training years | 17.25 ± 3.9 | 7.33 ± 4.7 | p < 0.001* |
| Training/week (hrs) | 6.28 ± 1.8 | 2.00 ± 1.4 | p < 0.001* |

All values are expressed as mean ± standard deviation. Group differences were tested with pairwise Mann-Whitney U tests.

*Significant differences between groups (p < 0.05).

## Sensorimotor skill learning: complex whole-body serial reaction time task (CWB-SRTT)

In this study, a four-directional CWB-SRTT for the lower extremities was used as a model of whole-body sensorimotor skill learning. The CWB-SRTT has been shown to robustly induce motor learning in two of our previous experiments [10, 11]. The main aspects of CWB-SRTT are illustrated in Fig 1, as well as briefly described below. For a detailed description of

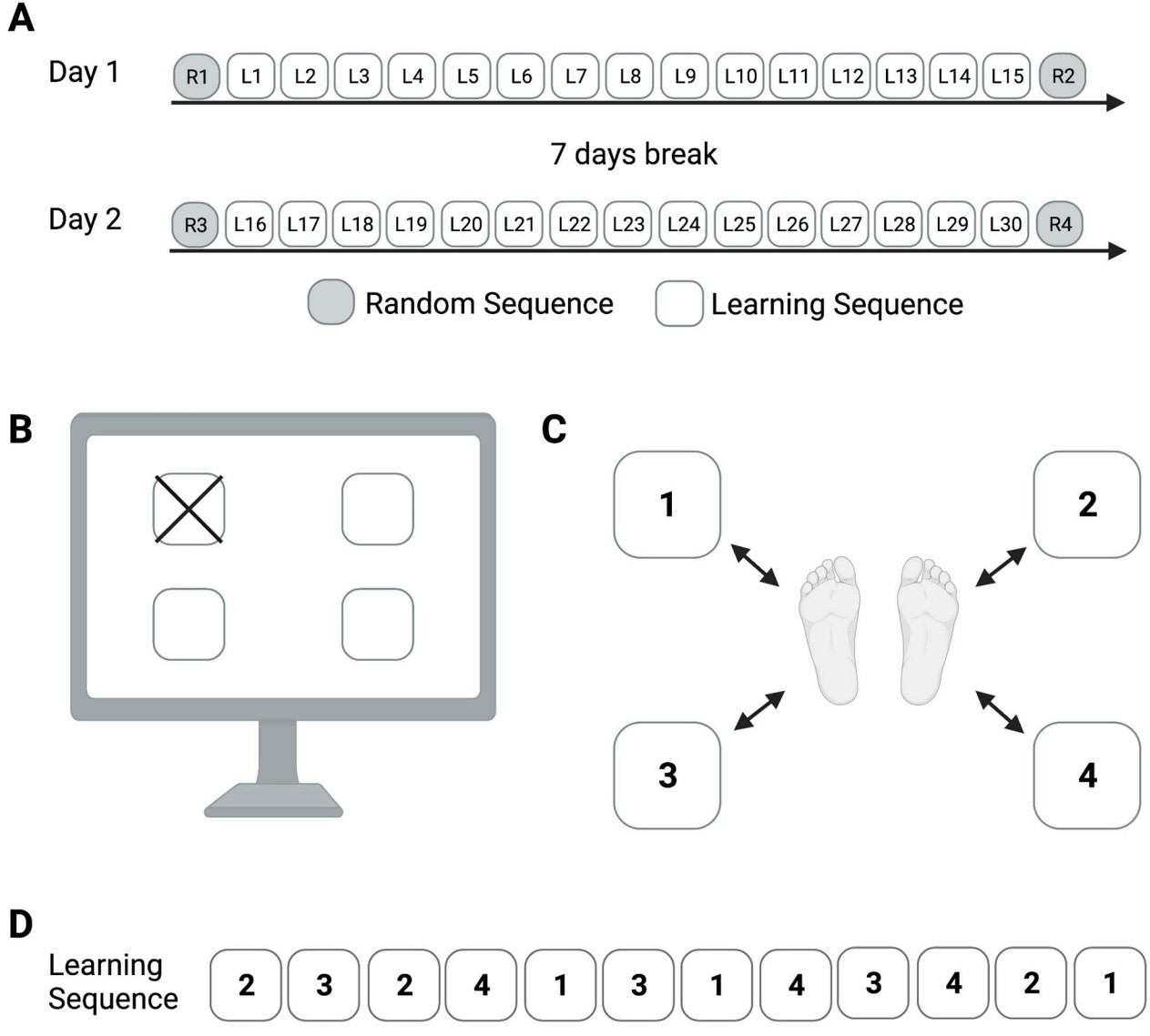

**Fig 1. Sensorimotor skill learning using a complex whole-body serial reaction time task (CWB-SRTT).** (A) A general overview of the study. Football players (FG) and non-athlete control participants (CG) completed CWB-SRTT on two days separated by a week. On each day participants performed 15 learning sequences (L1-L15; L16-L30), as well as one random sequence before and after the learning sequences (R1, R2, R3, R4). (B) A target cue displayed on any of four squares on a monitor situated 2 meters in front of the participant indicated the plate to be stepped on. (C) The participants' starting positions during the CWB-SRTT. The target plates were spaced by 0.5 m in both the lateral and longitudinal directions. All plates on the left side had to be operated with the left foot, and all plates on the right side had to be operated with the right foot. The number of plates in each position is related to a number in the learning sequence (1: front left; 2: front right; 3: back left; 4: back right). (D) The fixed learning sequence during all learning blocks appeared in the following order: 2-3-2-4-1-3-1-4-3-4-2-1. This figure was created with Biorender.com.

CWB-SRTT, please refer to Mizuguchi et al. (2019) [10]. In general terms, CWB-SRTT requires participants to step on one of four target plates as quickly as possible in response to visual stimuli. During performance, stimuli are presented in randomized or fixed orders to enable the analysis of sequence-specific and non-sequence-specific learning. For this study, we analyzed reaction time and movement time. The time between the onset of the visual stimulus and the raising of the response foot from one of the middle plates was used to determine reaction time. The time difference between lifting the response foot from the middle plate and making initial contact with one of the target plates was used to determine movement time. Median values of reaction and movement times were computed separately over each performed sequence block, resulting in 17 reaction and 17 movement times (2 random and 15 learning blocks) per experimental day and participant.

## Statistical analyses

All statistical analyses were performed using JASP (Version 0.16, JASP Team 2020). The normality of reaction and movement times was assessed and confirmed by Shapiro-Wilk testing ($\alpha = 0.05$). Demographic variables, Handedness/Footedness variables, and RECALL variables were not normally distributed. To compare these variables between FG and CG, non-parametric Mann-Whitney U tests were used.

To check whether the initial performance differed between FG and CG, reaction and movement times at the first random sequence on day 1 (R1) were compared using independent-sample t-tests.

For each experimental day, separate repeated-measures ANOVAs with the between-subject factor GROUP (FG, CG) and within-subject factor SEQUENCE (17 sequences) were used to evaluate sensorimotor skill learning within and between groups for reaction and movement times separately. A Greenhouse-Geisser correction was implemented when the sphericity assumption was violated.

On both days, the time difference between the last random sequence and the last learning sequence was calculated separately to assess sequence-specific improvements in reaction and movement times (day 1: R2-L15, day 2: R4-L30). Independent-sample t-tests were used to check for differences in learning rates between FG and CG.

The time difference between the first random sequence and the last random sequence on day 1 (R2-R1) and day 2 (R4-R3) was used to assess non-sequence-specific improvements in reaction and movement times separately. Again, these parameters were compared between groups using independent-sample t-tests.

Furthermore, we compared reaction times and movement times at the last learning sequence on day 1 (L15) and the first learning sequence on day 2 (L16) using a repeated-measures ANOVA with the between-subject factor GROUP (NAG, AG) and the within-subject factor SEQUENCE (L15, L16) to assess the retention of sequence-specific performance within and between groups.

We further computed the Spearman's rank correlation coefficients between the sequence-specific improvements on day 1 and RECALL1 as well as between sequence-specific improvements on day 2 and RECALL2 for reaction and movement times separately to assess if the sequence-specific improvement was related to the number of recalled items.

For the separate repeated-measures ANOVAs investigating reaction time or movement time on day 1 and day 2 the statistical threshold was Bonferroni adjusted to $p < 0.025$ to account for multiple comparisons. For all further analyses, the statistical threshold was set at $p < 0.05$. Effect sizes were expressed either using Cohen's d for t-tests, the rank-biserial correlation for Mann-Whitney U tests, or partial eta squared ($\eta_p^2$) for ANOVAs.

## Results

No significant differences in terms of age (W = 158.0, p = 0.134, $r_{biserial}$ = 0.317), handedness (W = 100.5, p = 0.440, $r_{biserial}$ = -0.163) and footedness (W = 73.0, p = 0.065, $r_{biserial}$ = -0.392) were observed between FG and CG. However, groups significantly differed in the number of training years (W = 229.0, p < 0.001, $r_{biserial}$ = 0.908) and the amount of training performed during an average week (W = 240.0, p < 0.001, $r_{biserial}$ = 1.000).

Initial performance in reaction times did not differ between FG and CG ($t_{(29)}$ = -1.371, p = 0.181, d = -0.493). Furthermore, initial movement times were not different between groups ($t_{(29)}$ = -1.508, p = 0.142, d = -0.542).

Repeated measures ANOVA indicated a significant effect for SEQUENCE ($F_{(6.196, 176.693)}$ = 9.915, p < 0.001, $\eta_p^2$ = .255) on median reaction times during day 1 (see Fig 2A). However, no

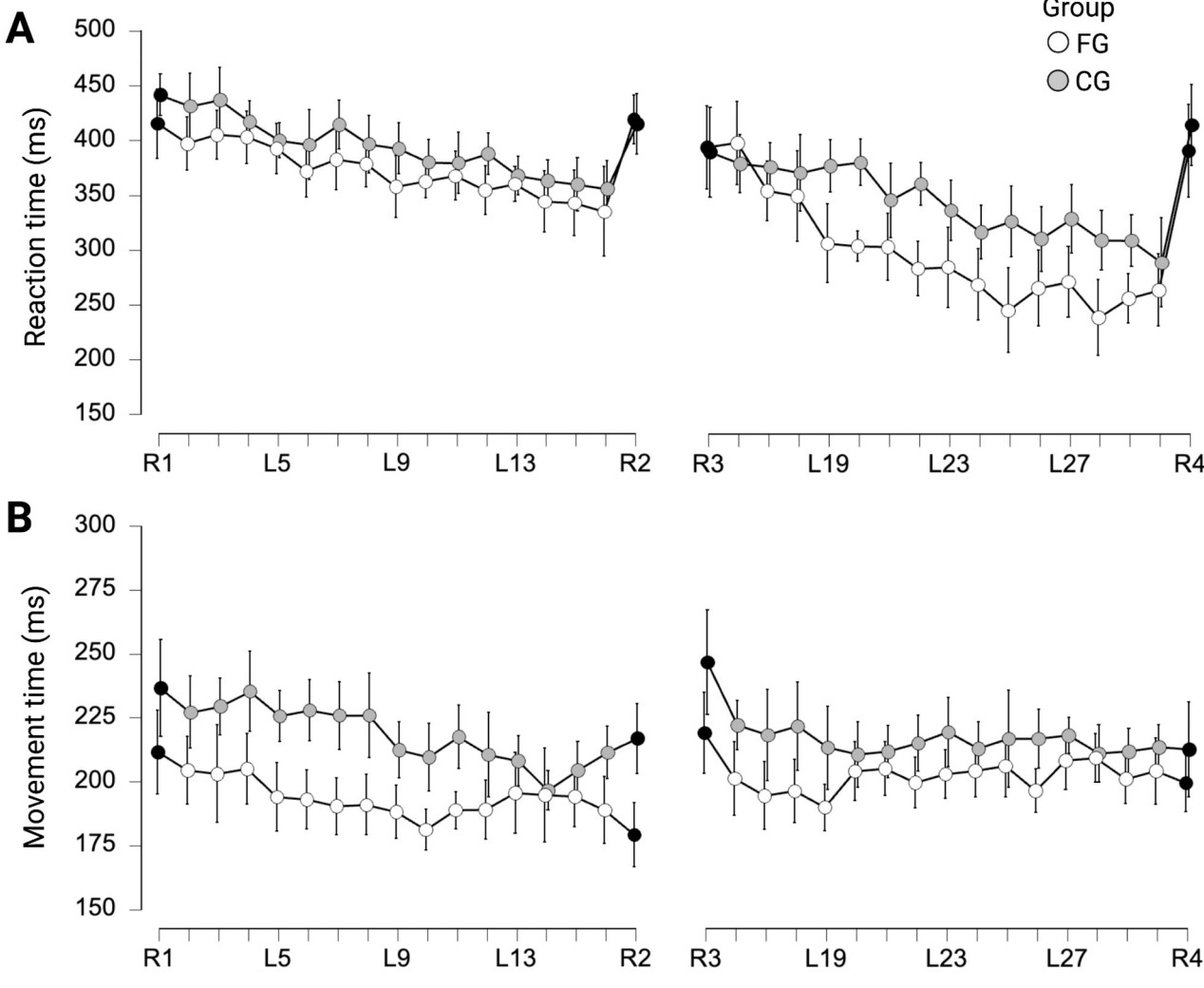

**Fig 2. Results of complex whole-body serial reaction time task (CWB-SRTT).** (A) For the control group (CG) and the football group (FG), a line graph showing CWB-SRTT learning for reaction times on day 1 and day 2 is shown. The average reaction times for each conducted sequence are displayed. The error bars represent the 95% confidence interval of the mean. Random sequences are indicated by black points. (B) A line graph depicting CWB-SRTT learning for movement times on day 1 and day 2 for the control group (CG) and the football group (FG). The mean movement times for each executed sequence are displayed. The error bars represent the 95% confidence interval of the mean. Random sequences are indicated by black points. This figure was created with Biorender.com.

significant effect was found for GROUP ($F_{(1, 29)}$ = 1.209, p = 0.281, $\eta_p^2$ = .040) and no significant interaction effect GROUP×SEQUENCE was observed ($F_{(6.196, 176.693)}$ = 0.451, p = 0.849, $\eta_p^2$ = .015).

A similar relation was observed for reaction times on day 2. A repeated-measures ANOVA revealed a significant effect for SEQUENCE ($F_{(5.752, 166.806)}$ = 16.546, p < 0.001, $\eta_p^2$ = .363), while no such effect could be observed for GROUP ($F_{(1, 29)}$ = 2.964, p = 0.096, $\eta_p^2$ = .093) and GROUP×SEQUENCE ($F_{(5.752, 166.806)}$ = 1.876, p = 0.091, $\eta_p^2$ = .061; see Fig 2A).

For movement times (see Fig 2B), repeated measures ANOVA revealed a significant effect for SEQUENCE on day 1 ($F_{(6.330, 183.569)}$ = 4.030, p < 0.001, $\eta_p^2$ = .122) and day 2 ($F_{(5.502, 159.562)}$ = 5.502, p = 0.001, $\eta_p^2$ = .078). Again, no significant effect was found for GROUP on day 1 ($F_{(1, 29)}$ = 2.477, p = 0.126, $\eta_p^2$ = .079) or day 2 ($F_{(1, 29)}$ = 0.987, p = 0.329, $\eta_p^2$ = .033) and no significant interaction effect GROUP×SEQUENCE was observed on day 1 ($F_{(6.330, 183.569)}$ = 1.235, p = 0.289, $\eta_p^2$ = .041) or day 2 ($F_{(5.502, 159.562)}$ = 0.839, p = 0.533, $\eta_p^2$ = .028).

For sequence specific improvement in reaction times, no differences were found on day 1 (84.00 ms vs. 59.33 ms; $t_{(29)}$ = 0.836, p = 0.410, d = 0.301) and on day 2 (125.94 ms vs. 124.33 ms; $t_{(29)}$ = 0.043, p = 0.966, d = 0.016) between FG and CG. Furthermore, no differences in non-sequence-specific reaction time improvement were found between FG and CG on day 1 (3.78 ms vs. -26.47 ms; $t_{(29)}$ = 1.371, p = 0.181, d = 0.493) and day 2 (-3.13 ms vs. 24.40 ms; $t_{(29)}$ = -1.150, p = 0.259, d = -0.413).

Regarding sequence specific improvement in movement times, no differences were found on day 1 (-9.63 ms vs. 5.50 ms; $t_{(29)}$ = -1.855, p = 0.074, d = -0.667) and on day 2 (-4.72 ms vs. -0.90 ms; $t_{(29)}$ = -0.314, p = 0.756, d = -0.113) between FG and CG. Furthermore, no differences in non-sequence-specific movement time improvement were found between FG and CG on day 1 (-32.41 ms vs. -19.67 ms; $t_{(29)}$ = -0.859, p = 0.397, d = -0.309) and day 2 (-19.47 ms vs. -33.83 ms; $t_{(29)}$ = 1.039, p = 0.307, d = 0.373).

In terms of retention of sequence-specific performance in reaction times from day 1 to day 2, repeated-measures ANOVA revealed a significant effect for SEQUENCE ($F_{(1, 29)}$ = 7.510, p = 0.010, $\eta_p^2$ = .206). Post-hoc comparison showed that reaction times during L16 on day 2 were significantly higher compared to reaction times during L15 on day 1 (mean difference (MD) = -43.89 ms, SE = 16.02, p = 0.010, d = -.492). However, there was no significant effect for GROUP ($F_{(1, 29)}$ = 0.146, p = 0.706, $\eta_p^2$ = .005) and no significant interaction effect GROUP×SEQUENCE ($F_{(1, 29)}$ = 0.605, p = 0.443, $\eta_p^2$ = .020), indicating that groups showed no difference in the degree of retention.

A similar relation was observed for retention of sequence-specific performance in movement times from day 1 to day 2. Repeated-measures ANOVA revealed a significant effect for SEQUENCE ($F_{(1, 29)}$ = 21.254, p < 0.001, $\eta_p^2$ = .423). Post-hoc comparison showed that reaction times during L16 on day 2 were significantly higher compared to response times during L15 on day 1 (mean difference (MD) = -31.82 ms, SE = 6.90, p < 0.001, d = -.828). No significant effect for GROUP ($F_{(1, 29)}$ = 2.817, p = 0.104, $\eta_p^2$ = .089) and no significant interaction effect GROUP×SEQUENCE ($F_{(1, 29)}$ = 0.119, p = 0.733, $\eta_p^2$ = .004) was observed, again indicating that groups showed no difference in the degree of retention.

FB and CG did not differ in the number of correctly recalled sequence items on day 1 (W = 152.5, p = 0.200, $r_{biserial}$ = 0.271) or day 2 (W = 127.5.0, p = 0.775, $r_{biserial}$ = 0.063). Interestingly however, Spearman rank correlation between sequence-specific improvements in reaction times on day 1 and RECALL1 ($r_s$ = 0.465, p = 0.008, 95% CI [0.133, 0.703]) as well as sequence-specific improvements in reaction times on day 2 an RECALL2 ($r_s$ = 0.698, p < 0.001, 95% CI [0.177, 0.726]) showed a significant positive relationship when both groups were pooled (see Fig 3). When both groups were separated, FG did not show a significant

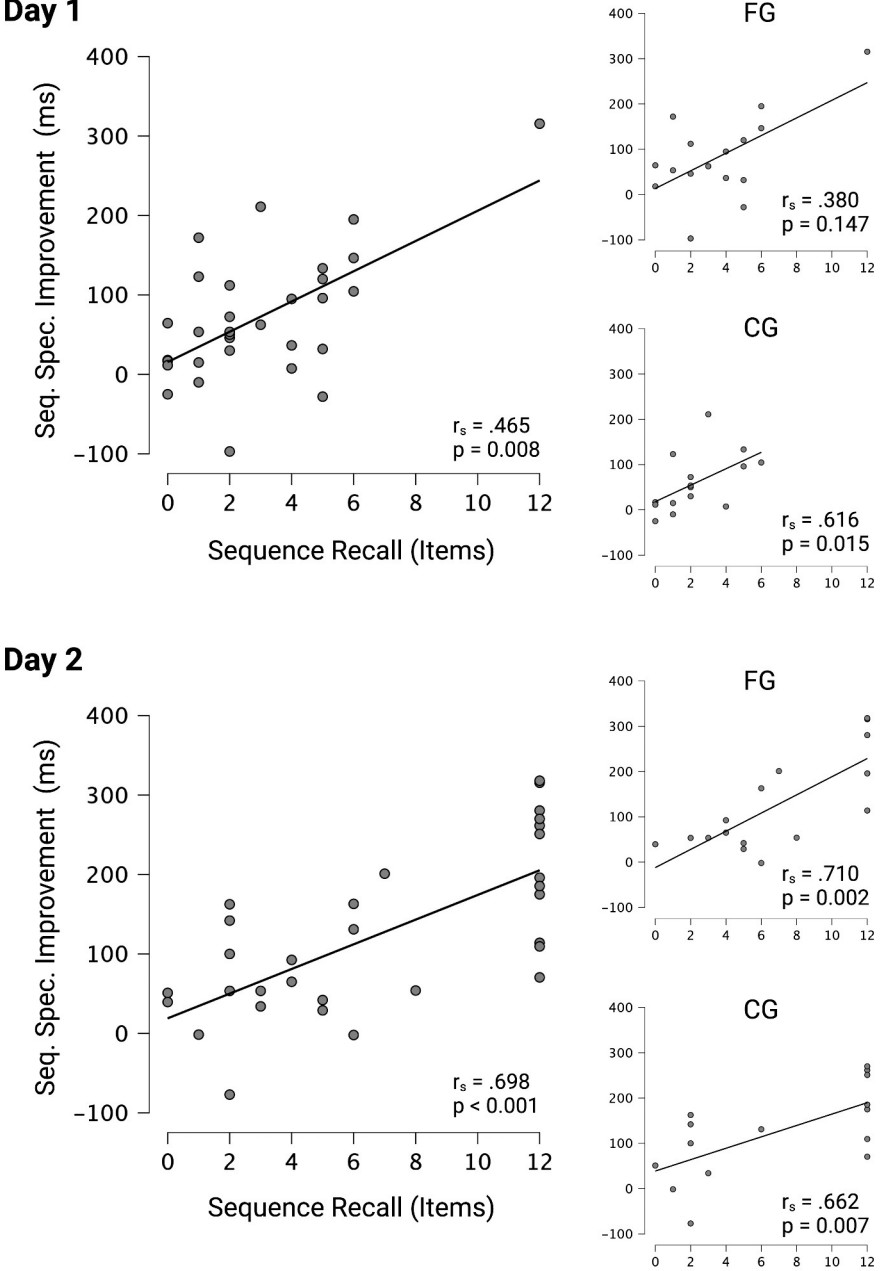

**Fig 3. Positive association between the number of correctly recalled items of the CWB-SRTT learning sequence and sequence-specific improvement in reaction times.** Spearman rank-correlation between sequence recall and sequence-specific improvement in reaction times for the pooled sample on day 1 and day 2. Furthermore, these correlations were calculated separately for FG and CG on each learning day. This figure was created with Biorender. com.

correlation on day 1 ($r_s$ = 0.380, p = 0.147, 95% CI [-0.143, 0.737]) but on day 2 ($r_s$ = 0.710, p = 0.002, 95% CI [0.330, 0.892]), whereas CG showed significant correlations both on day 1 ($r_s$ = 0.616, p = 0.0015, 95% CI [0.151, 0.858]) and day 2 ($r_s$ = 0.662, p = 0.007, 95% CI [0.226, 0.877]).

However, this observation could not be confirmed for movement times on day 1 ($r_s$ = -0.025, p = 0.894, 95% CI [-0.376, 0.332]) or day 2 ($r_s$ = -0.115, p = 0.537, 95% CI [0.177, 0.726]).

## Discussion

In the present study, we aimed to investigate potential motor transfer effects on complex serial reaction time task performance by comparing football players (FG) and non-athletes (CG) in their ability to perform and learn a complex whole-body serial reaction time task (CWB-SRTT). We analyzed both movement and reaction times separately to disentangle potential modulations underlying CWB-SRTT performance. When comparing CG and FG, initial reaction and movement times did not differ significantly between groups. Furthermore, there were no differences in sequence-specific or non-sequence-specific improvements between CG with FG. Analysis of learning rates between CG and FG did not reveal any significant differences between groups at either day 1 or day 2 for both movement and reaction time. Although we found a significant effect for SEQUENCE on retention rates of movement and reaction times, we were unable to demonstrate a significant interaction between GROUP and SEQUENCE. These results suggest that there were no significant differences in retention rates between groups. FG and CG did not differ in the number of correctly recalled test items. However, when pooling FG and CG, correlation analysis revealed a significant positive correlation between the number of correctly recalled items and sequence-specific improvement in reaction times but not movement times on both training days.

Given the similarity between football-typical movement patterns and motor requirements of the CWB-SRTT [12], we expected better initial performance of FG. Although FG reaction and movement times were faster compared to CG, these differences were not significant. Accordingly, we did not observe motor transfer effects of football-specific performance on initial CWB-SRTT performance. In general, athletes of various sports disciplines exhibit better reaction times, when compared to non-athletes [17, 18]. However, and in line with the present findings, such differences are closely related to task familiarity [19, 20]. Therefore, although the similarities between football-specific movement patterns and the CWB-SRTT are present in terms of rapid motor actions and reactions of lower extremities, the requirements potentially differ in terms of visuomotor information integration [10]. For this reason, a possible explanation for the lack of significant differences in initial performance is that both groups were naïve to the task. CWB-SRTT does contain motor elements that resemble those of football. Nevertheless, the task performed here is not a typical component of football training regimes. One might argue that the level of expertise plays a substantial role in potential motor transfer effects on initial CWB-SRTT performance. We consider this unlikely since it was shown that football players with different competition levels do not significantly differ in reaction times [21]. Contrasting results exist, but only concerning reaction time in general motor inhibition tasks [13]. On average, reaction and movement times of FG were ~25 ms faster compared to CG. Both reaction and movement times have a considerable impact on athletic performance, as both measures are closely related to overall sensorimotor capacity [22]. Therefore, it is not surprising that FG showed faster reaction and movement times compared to CG. An additional factor that might affect motor transfer is the variability of training schedules [23]. Training variability has previously been shown to have a positive effect on motor transfer in motor sequence learning tasks [24]. However, it is difficult to retrospectively quantify the degree of variability in training regimes of athletes with extensive training backgrounds. In our study, CG trained nonspecifically for 7 years at a maximum workload of 2 hours per week whereas FG trained football-specifically for 17 years at a workload of 6.2 hours per week. In future studies, it seems to be of interest to modulate the variability of a specific

training program within longitudinal designs to uncover the influence of variability on motor transfer.

Both groups improved their movement times and reaction times during CWB-SRTT on day 1 and day 2. This is in line with previous findings for simple SRTT paradigms (Moisello et al., 2009) [8] as well as for CWB-SRTT (Maudrich et al., 2021) [11]. Similar to our study comparing strength and endurance athletes to healthy controls during CWB-SRTT, we did not detect significant differences in sequence-specific improvements when comparing movement times and reaction times between both groups on either day. Therefore, and contrary to our hypothesis, football players do not show significantly better learning rates in CWB-SRTT compared to non-athletes. Sequence-specific learning is closely associated with the amount of accurately recalled items within the motor sequence [23]. Knowledge of the sequence of items improves response times, as the associated increases in declarative knowledge lead to better anticipation of subsequent stimuli [25]. In our study, both groups did not differ in the amount of correctly recalled items. Interestingly, correlation analyses revealed a significant relationship between the number of correctly recalled items and sequence-specific improvements in reaction time, but not movement time, when both groups were pooled together. An exception was FG, which failed to show this relationship on day 1, but did so on day 2. Based on these results, it can be assumed that improvements in reaction times observed in both groups are predominately related to the proficiency of item recall. Since such a relationship could not be replicated for movement times, we speculate that the observed improvements are predominately cognitive. Sequence-specific improvements in participants might therefore reflect adaptations in the cognitive domain rather than adaptations in the motor domain [10]. Although the initially lower reaction and movement times suggest an inherent motor advantage of FG, this may have been negated by predominantly cognitive-induced improvements during learning of CWB-SRTT as observed in this study. This could account for the lack of differences in sequence-specific improvements between groups as, although the motor overlap between CWB-SRTT and football-specific movement patterns is high, the cognitive overlap, i.e., stimuli-response relationships is potentially low. Recent results demonstrate that non-sport-specific training in a visuomotor task improves cognitive but not sport-specific motor performance within such tasks [26]. Furthermore, unspecific training of general motor skills appears to improve complex motor performance in football players compared to sport-specific training [27]. Thus, despite the fact that gross motor skills are an integral part of sport-specific skill development [28, 29], the environment in which such skills are acquired is crucial to the development of adaptive strategies for adequate regulation of perception and action [26, 30].

For non-sequence-specific learning, we did not find differences between CG and FG. Similar to previous research, this may be caused by the structure of sequence and random blocks within the CWB-SRTT. While participants complete multiple identical sequence blocks, the possibility of a transfer from implicit to explicit learning strategies might increase [23]. As implicit learning has been demonstrated to be more effective concerning motor transfer to a novel sequence when compared to explicit learning [31], it is an important aspect to monitor during SRTT. However, no differences were found concerning the number of recalled items on day 1 and day 2 between groups. Therefore, implicit and explicit learners were equally distributed between CG and FG. It seems of value to monitor sequence recall after each block to obtain the time point at which correct item recall is present for the first time, although this might inadvertently induce explicit learning. In any case, future studies should address this aspect to better disentangle the underlying processes of CWB-SRTT learning.

## Conclusion

With this study, we extend previous findings on motor transfer effects by showing that football players and non-athletes show no differences in their ability to learn a novel motor sequence using CWB-SRTT. Although faster reaction and movement times were observed among football players compared to non-athletes prior to training, differences between groups were not statistically significant. Furthermore, sequence-specific and non-sequence-specific improvements after learning did not differ between groups. We hypothesize that this might be due to the fact that there did not appear to be any substantial transfer of cognitive-motor football skill to learning the novel movement sequence of the CWB-SRTT. The study of motor transfer effects in complex whole-body movements is important for both recreational and competitive sports. It is often beneficial when a newly learned skill can be applied outside of the specific context in which it was acquired. Furthermore, physical education still wavers between early specialization and general education. In this context, investigating the scope of validity as well as the determinants of motor transfer seems to be a promising approach. Future studies should attempt to extend our research to different tasks and athlete populations to uncover possible principles and mechanisms of motor transfer effects.

## Acknowledgments

We acknowledge support from Leipzig University for Open Access Publishing. Furthermore, we thank Hartmut Domröse for technical support.

## Author Contributions

**Conceptualization:** Tom Maudrich, Patrick Ragert, Rouven Kenville.

**Data curation:** Hannah Kandt.

**Investigation:** Rouven Kenville.

**Methodology:** Tom Maudrich, Patrick Ragert.

**Resources:** Patrick Ragert.

**Supervision:** Patrick Ragert, Rouven Kenville.

**Visualization:** Tom Maudrich.

**Writing – original draft:** Tom Maudrich, Rouven Kenville.

**Writing – review & editing:** Tom Maudrich, Hannah Kandt, Patrick Ragert, Rouven Kenville.

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
