## [Decision Letter · Decision Letter 0]

31 May 2022

PONE-D-22-02272Whole-Body Sensorimotor Skill Learning in Football Players: No Evidence for Motor Transfer EffectsPLOS ONE

Dear Dr. Maudrich,

Thank you for submitting your manuscript to PLOS ONE. After careful consideration, we feel that it has merit but does not fully meet PLOS ONE’s publication criteria as it currently stands. Therefore, we invite you to submit a revised version of the manuscript that addresses the points raised during the review process.

The paper is well structured and describes a well conducted study. There are a few points that could be explaine in more detail, but generally the paper is publishable.

One suggestion would be a more detailled description of the control group. What sport were they doing and on what level?

Please also address the points of the reviewer carefully.

We look forward to receiving your revised manuscript.

Kind regards,

Peter Andreas Federolf

Academic Editor

PLOS ONE

Journal Requirements:

2. Please note that in order to use the direct billing option the corresponding author must be affiliated with the chosen institute. Please either amend your manuscript to change the affiliation or corresponding author, or email us at plosone@plos.org with a request to remove this option.

Additional Editor Comments:

Dear authors,

I approached 21 potential reviewers, but found only one willing to provide an assessment of the paper. Therefore I took a careful look at the manuscript myself and feel confident to go forward with only 1 review. The paper is well structured and describes a well conducted study. There are a few points that could be explaine in more detail, but generally the paper is publishable.

One suggestion I have, would be a more detailled description of the control group. What sport were they doing and on what level?

Please also address the points of the other reviewer carefully.

Reviewers' comments:

Reviewer's Responses to Questions

**Comments to the Author**

1. Is the manuscript technically sound, and do the data support the conclusions?

Reviewer #1: Yes

2. Has the statistical analysis been performed appropriately and rigorously? 

Reviewer #1: Yes

3. Have the authors made all data underlying the findings in their manuscript fully available?

Reviewer #1: Yes

4. Is the manuscript presented in an intelligible fashion and written in standard English?

Reviewer #1: No

5. Review Comments to the Author

Reviewer #1: Specific comments, suggestions, and questions are referenced to manuscript line numbers:

27-28: Does the phrase “increasing the overlap of motor demands between CWB-SRTT and an athlete group” mean that you were attempting to more closely match the demands of the CWB-SRTT with those imposed by participation in football (i.e., soccer)? If so, “motor demands” are characteristics of the sport activity. The “athlete group” does not possess motor demands.

33: Replace “as well as” with “nor” for consistency with “did not reveal any differences…”

39: The term “motor transfer” needs to be more clearly defined. An “athlete advantage over non-athletes when learning” does not define motor transfer.

49-50: The first sentence of the manuscript seems incomplete and is a bit unclear. Don’t you mean “the extent to which specific athlete performance capabilities exhibited during participation in a given sport-related activity translate to superior motor performance in a somewhat different activity”? The content of the sentence that follows seems to indicate that is what you intend to convey.

64-65: The extent to which a person has learned a motor sequence could be determined without consideration of response speed (i.e., proportion of correct motor responses). Are you suggesting that faster reaction is an indicator of superior motor sequence learning?

85-86: Redundant content. The is no need to repeat the information presented in prior text.

90: “Athletes from several sports can be considered as potential study populations…”

91-93, 100-101: Football (i.e., soccer) involves execution of motor sequences that are highly unpredictable, which leads to implicit learning of innumerable combinations, rather than explicit learning of specific sequences. This distinction between football performance in a competitive environment and th27-28: Does the phrase “increasing the overlap of motor demands between CWB-SRTT and an athlete group” mean that you were attempting to more closely match the demands of the CWB-SRTT with those imposed by participation in football (i.e., soccer)? If so, “motor demands” are characteristics of the sport activity. The “athlete group” does not possess motor demands.

33: Replace “as well as” with “nor” for consistency with “did not reveal any differences…”

39: The term “motor transfer” needs to be more clearly defined. An “athlete advantage over non-athletes when learning” does not define motor transfer.

49-50: The first sentence of the manuscript seems incomplete and is a bit unclear. Don’t you mean “the extent to which specific athlete performance capabilities exhibited during participation in a given sport-related activity translate to superior motor performance in a somewhat different activity”? The content of the sentence that follows seems to indicate that is what you intend to convey.

64-65: The extent to which a person has learned a motor sequence could be determined without consideration of response speed (i.e., proportion of correct motor e nature of the CWB-SRTT needs to be emphasized. Furthermore, references to “motor actions, motor transfer, and motor skill” fail to acknowledge the complex “cognitive-motor” integration process that is required for efficient performance of whole-body movement sequences.

147-160: Apparently, 12 movements were performed during each “block” (Lines 129-131), which are also referred to as “learning sequences” and “random sequences” in the Figure 1 legend. I interpret this to mean that a total of 204 movements were performed during the Day 1 and Day 2 sessions. This information should be provided.

171: Why wasn’t a Bonferroni alpha-level correction used for multiple comparisons (i.e., 2 separate ANOVAs for reaction time and movement time)?

280: Delete the word “Both” at the beginning of the sentence.

284: I suggest replacing the phrase “comparably large intersection between” with “similarity between…”

286: I suggest replacing the word “lower” (which could be interpreted as a negative finding) with “faster…”

294: Specify “a possible explanation for the lack of significant differences…”

304: I suggest replacing the word “lower” (which could be interpreted as a negative finding) with “faster…”

318-319: The preceding content did not clearly provide an explanation for why you expected football players to demonstrate better learning rates for than non-athletes for a specific sequence of movement cues. I fail to see the relevance of sequence-specific learning to the high degree of uncertainty about the complex motor actions that will need to be rapidly executed in a competitive sport environment.

330-337: I agree that the sequence-specific improvements probably represent a cognitive adaptation in memory processes, but “sensorimotor processes” are not distinct from cognitive processes. The content of this sentence implies that the cognitive domain is distinct from the sensorimotor domain.

355: I suggest replacing the phrase “In any way” with “In any case…”

360-361: The content of this sentence is very confusing. Suggested rewording: “Although faster reaction and movement times were observed among football players compared to non-athletes prior to training, differences between groups were not statistically significant for either random or sequence-specific movement cues.”

362-363: Suggested rewording for greater clarity: “There did not appear to be any substantial transfer of cognitive-motor football skill to learning the novel movement sequences of the CWB-SRTT.”

6. PLOS authors have the option to publish the peer review history of their article (what does this mean?). If published, this will include your full peer review and any attached files.

Reviewer #1: **Yes: **Gary B. Wilkerson

---

## [Author Response · Author response to Decision Letter 0]

3 Jun 2022

Response to Reviewers

Additional Editor Comments:

Dear authors,

I approached 21 potential reviewers, but found only one willing to provide an assessment of the paper. Therefore I took a careful look at the manuscript myself and feel confident to go forward with only 1 review. The paper is well structured and describes a well conducted study. There are a few points that could be explained in more detail, but generally the paper is publishable.

One suggestion I have, would be a more detailed description of the control group. What sport were they doing and on what level?

Our response: Thank you for this suggestion. We added this information:

“Non-athletes did not participate in a specific sport in an organized manner apart from general recreational activities (e.g., running, fitness and cycling).” (lines 114-115)

Please also address the points of the other reviewer carefully.

Reviewers' comments:

Reviewer #1: Specific comments, suggestions, and questions are referenced to manuscript line numbers:

27-28: Does the phrase “increasing the overlap of motor demands between CWB-SRTT and an athlete group” mean that you were attempting to more closely match the demands of the CWB-SRTT with those imposed by participation in football (i.e., soccer)? If so, “motor demands” are characteristics of the sport activity. The “athlete group” does not possess motor demands.

Our response: Thank you for pointing this out. We changed the sentence accordingly.

“In the present study, we aimed to extend this research by increasing the overlap of task requirements between CWB-SRTT and a specific sports discipline.” (lines 25-26)

33: Replace “as well as” with “nor” for consistency with “did not reveal any differences…”

Our response: Done.

39: The term “motor transfer” needs to be more clearly defined. An “athlete advantage over non-athletes when learning” does not define motor transfer.

Our response: You are correct. We defined motor transfer more clearly.

“Motor transfer refers to the transferability of a specific motor skill to different motor tasks” (line 38)

49-50: The first sentence of the manuscript seems incomplete and is a bit unclear. Don’t you mean “the extent to which specific athlete performance capabilities exhibited during participation in a given sport-related activity translate to superior motor performance in a somewhat different activity”? The content of the sentence that follows seems to indicate that is what you intend to convey.

Our response: Again, you are correct. Your suggestion makes this part clearer.

“There is an ongoing debate about the extent to which specific athletic performance capabilities exhibited during participation in a given sport-related activity translate to superior motor performance in an unrelated motor scenario.” (lines 47-49)

64-65: The extent to which a person has learned a motor sequence could be determined without consideration of response speed (i.e., proportion of correct motor responses). Are you suggesting that faster reaction is an indicator of superior motor sequence learning?

64-65: The extent to which a person has learned a motor sequence could be determined without consideration of response speed (i.e., proportion of correct motor e nature of the CWB-SRTT needs to be emphasized. Furthermore, references to “motor actions, motor transfer, and motor skill” fail to acknowledge the complex “cognitive-motor” integration process that is required for efficient performance of whole-body movement sequences.

Our response: Thank you for this comment. Indeed, faster reaction times is not the only indicator of motor sequence learning. Successful sequence learning is also determined by the proportion of correct motor responses. We refined the statement accordingly.

“During SRTT, participants are presented with spatially separated visual stimuli. Participants must respond as accurately and as quickly as possible with a motor action, e.g., pressing a button, corresponding to the sequence of the presented stimuli. Performance of motor sequence learning is measured by the proportion of correct responses and the reaction time to the presented visual cues. Consequently, SRTT performance relies on the successful integration of motor and cognitive processes.” (lines 62-67)

85-86: Redundant content. The is no need to repeat the information presented in prior text.

Our response: We deleted this redundancy.

90: “Athletes from several sports can be considered as potential study populations…”

Our response: Changes were made accordingly.

91-93, 100-101: Football (i.e., soccer) involves execution of motor sequences that are highly unpredictable, which leads to implicit learning of innumerable combinations, rather than explicit learning of specific sequences. This distinction between football performance in a competitive environment and th

Our response: Unfortunately, parts of this comment are missing in the revision letter. However, we think that this comment relates to another comment of yours (318-319: The preceding content did not clearly provide an explanation for why you expected football players to demonstrate better learning rates for than non-athletes for a specific sequence of movement cues. I fail to see the relevance of sequence-specific learning to the high degree of uncertainty about the complex motor actions that will need to be rapidly executed in a competitive sport environment.). We hope that our answer to the latter comment (318-319) also clarifies concerns raised here.

147-160: Apparently, 12 movements were performed during each “block” (Lines 129-131), which are also referred to as “learning sequences” and “random sequences” in the Figure 1 legend. I interpret this to mean that a total of 204 movements were performed during the Day 1 and Day 2 sessions. This information should be provided.

Our response: This information was added to the experimental procedure.

“FG and CG completed 15 consecutive sequence blocks and one random block before and one random block after all sequence blocks on each day (Figure 1A). This leads to a total of 204 motor responses for each experimental day.” (lines 128-130)

171: Why wasn’t a Bonferroni alpha-level correction used for multiple comparisons (i.e., 2 separate ANOVAs for reaction time and movement time)?

Our response: You are correct, this has to be done. We corrected this point in the manuscript. However, all results remain the same.

“For the separate repeated-measures ANOVAs investigating reaction time or movement time on day 1 and day 2 the statistical threshold was Bonferroni adjusted to p < 0.025 to account for multiple comparisons.” (lines 190-192)

280: Delete the word “Both” at the beginning of the sentence.

Our response: Done.

284: I suggest replacing the phrase “comparably large intersection between” with “similarity between…”

Our response: This was done.

286: I suggest replacing the word “lower” (which could be interpreted as a negative finding) with “faster…”

Our response: Done. 

294: Specify “a possible explanation for the lack of significant differences…”

Our response: Done.

304: I suggest replacing the word “lower” (which could be interpreted as a negative finding) with “faster…”

Our response: Done.

318-319: The preceding content did not clearly provide an explanation for why you expected football players to demonstrate better learning rates for than non-athletes for a specific sequence of movement cues. I fail to see the relevance of sequence-specific learning to the high degree of uncertainty about the complex motor actions that will need to be rapidly executed in a competitive sport environment.

Our response: You are correct that sequence-specific learning and complex, unpredictable sporting demands such as those in football initially appear unrelated. 

However, the initial challenges of CWB-SRTT are also classified as "unpredictable" up to the point of recognizing a sequence, i.e., the transfer from implicit to explicit learning (Robertson, 2007). The idea behind our hypothesis derives rather from the overlap of the motor component of the task, i.e., the rapid, goal-directed activation of the lower extremities. Especially due to the unpredictability in the football context, football players show a superior repertoire of movement patterns and thus motor ability in this point, which, as hypothesized, could manifest itself in a positive learning performance within the CWB-SRTT. We specified this point in the manuscript where we mention our hypothesis of the study.

“In football, offensive, as well as defensive game decisions, are realized in the range of a few hundredths of a second via motor actions of the lower extremities (Vijayendra and Neelam, 2020). Accordingly, reaction time is an important determinant of football performance (Vijayendra and Neelam, 2020). Several studies illustrate this by demonstrating that the cognitive processing of soccer players, in the form of improved reaction times during general motor inhibition tasks (Verburgh et al., 2014) as well as faster stimulus processing times and improved attentional performance (Vestberg et al., 2017), is increased compared to control groups. Furthermore, the motor related overlap between CWB-SRTT and football performance, i.e., the rapid, goal-directed activation of the lower extremities, is considerably high. In sum, football players might therefore hold an advantage in learning a novel sensorimotor task such as the CWB-SRTT.” 

Consequently, the purpose of this study was to examine the difference between football players and non-athletes during CWB-SRTT performance on two separate days to investigate potential motor transfer of football skills when learning a new motor skill using CWB-SRTT. Based on the outlined body of research, we hypothesized improved initial performance as well as improved learning and retention rates for football players compared to non-athletes.“ (lines 88-101)

330-337: I agree that the sequence-specific improvements probably represent a cognitive adaptation in memory processes, but “sensorimotor processes” are not distinct from cognitive processes. The content of this sentence implies that the cognitive domain is distinct from the sensorimotor domain.

Our response: We agree that the term „sensorimotor domain” is slightly misleading as cognitive processes are implied in this term. Therefore, to more accurately describe our findings we changed it to “motor domain” in this specific passage. Our line of argumentation relates to the fact that the motor component, i.e., movement patterns of the lower extremity, shows considerable similarity concerning CWB-SRTT and football performance. As you correctly pointed out the cognitive demands of CWB-SRTT and football in a competitive sports environment differ.

“Sequence-specific improvements in participants might therefore reflect adaptations in the cognitive domain rather than adaptations in the motor domain (Mizuguchi et al., 2019). Although the initially lower reaction and movement times suggest an inherent motor advantage of FG, this may have been negated by predominantly cognitive-induced improvements during learning of CWB-SRTT as observed in this study. This could account for the lack of differences in sequence-specific improvements between groups as, although the motor overlap between CWB-SRTT and football-specific movement patterns is high, the cognitive overlap, i.e., stimuli-response relationships is potentially low.” (lines 330-337)

355: I suggest replacing the phrase “In any way” with “In any case…”

Our response: Done.

360-361: The content of this sentence is very confusing. Suggested rewording: “Although faster reaction and movement times were observed among football players compared to non-athletes prior to training, differences between groups were not statistically significant for either random or sequence-specific movement cues.”

Our response: Thank you for the suggestion. We used it to improve the wording of this sentence.

„Although faster reaction and movement times were observed among football players compared to non-athletes prior to training, differences between groups were not statistically significant. Furthermore, sequence-specific and non-sequence-specific improvements after learning did not differ between groups.” (lines 358-361)

362-363: Suggested rewording for greater clarity: “There did not appear to be any substantial transfer of cognitive-motor football skill to learning the novel movement sequences of the CWB-SRTT.”

Our response: Thank you. We changed the sentence accordingly.

“We hypothesize that this might be due to the fact that there did not appear to be any substantial transfer of cognitive-motor football skill to learning the novel movement sequence of the CWB-SRTT.” (lines 361-363)

---

## [Decision Letter · Decision Letter 1]

30 Jun 2022

Whole-body sensorimotor skill learning in football players: No evidence for motor transfer effects

PONE-D-22-02272R1

Dear Dr. Maudrich,

We’re pleased to inform you that your manuscript has been judged scientifically suitable for publication and will be formally accepted for publication once it meets all outstanding technical requirements.

Kind regards,

Peter Andreas Federolf

Academic Editor

PLOS ONE

Additional Editor Comments (optional):

Reviewers' comments:

Reviewer's Responses to Questions

**Comments to the Author**

1. If the authors have adequately addressed your comments raised in a previous round of review and you feel that this manuscript is now acceptable for publication, you may indicate that here to bypass the “Comments to the Author” section, enter your conflict of interest statement in the “Confidential to Editor” section, and submit your "Accept" recommendation.

Reviewer #1: All comments have been addressed

2. Is the manuscript technically sound, and do the data support the conclusions?

Reviewer #1: Yes

3. Has the statistical analysis been performed appropriately and rigorously? 

Reviewer #1: Yes

4. Have the authors made all data underlying the findings in their manuscript fully available?

Reviewer #1: Yes

5. Is the manuscript presented in an intelligible fashion and written in standard English?

Reviewer #1: Yes

6. Review Comments to the Author

Reviewer #1: All prior review comments have been satisfactorily addressed by the authors. The manuscript provides a very thorough presentation of the study methods, results, and clinical applications.

7. PLOS authors have the option to publish the peer review history of their article (what does this mean?). If published, this will include your full peer review and any attached files.

Reviewer #1: **Yes: **Gary B. Wilkerson, EdD, ATC

---

## [Editor Report · Acceptance letter]

4 Jul 2022

PONE-D-22-02272R1 

Whole-body sensorimotor skill learning in football players: No evidence for motor transfer effects 

Dear Dr. Maudrich:

I'm pleased to inform you that your manuscript has been deemed suitable for publication in PLOS ONE. Congratulations! Your manuscript is now with our production department. 

Kind regards, 

on behalf of

Dr. Peter Andreas Federolf 

Academic Editor

PLOS ONE